# Constant Approximation for Individual Preference Stable Clustering

**Anders Aamand**
MIT
aamand@mit.edu

**Justin Y. Chen**
MIT
justc@mit.edu

**Allen Li**
MIT
cliu568@mit.edu

**Sandeep Silwal**
MIT
silwal@mit.edu

**Pattara Sukprasert**[*]
Databricks
pat.sukprasert@databricks.com

**Ali Vakilian**
TTIC
vakilian@ttic.edu

**Fred Zhang**
UC Berkeley
z0@berkeley.edu

## Abstract

Individual preference (IP) stability, introduced by Ahmadi et al. (ICML 2022), is a natural clustering objective inspired by stability and fairness constraints. A clustering is $\alpha$-IP stable if the average distance of every data point to its own cluster is at most $\alpha$ times the average distance to any other cluster. Unfortunately, determining if a dataset admits a 1-IP stable clustering is NP-Hard. Moreover, before this work, it was unknown if an $o(n)$-IP stable clustering always *exists*, as the prior state of the art only guaranteed an $O(n)$-IP stable clustering. We close this gap in understanding and show that an $O(1)$-IP stable clustering always exists for general metrics, and we give an efficient algorithm which outputs such a clustering. We also introduce generalizations of IP stability beyond average distance and give efficient near optimal algorithms in the cases where we consider the maximum and minimum distances within and between clusters.

## 1 Introduction

In applications involving and affecting people, socioeconomic concepts such as game theory, stability, and fairness are important considerations in algorithm design. Within this context, Ahmadi et al. [1] (ICML 2022) introduced the notion of *individual preference stability (IP stability)* for clustering. At a high-level, a clustering of an input dataset is called 1-IP stable if, for each individual point, its average distance to any other cluster is larger than the average distance to its own cluster. Intuitively, each individual prefers its own cluster to any other, and so the clustering is stable.

There are plenty of applications of clustering in which the utility of each individual in any cluster is determined according to the other individuals who belong to the same cluster. For example, in designing *personalized medicine*, the more similar the individuals in each cluster are, the more effective medical decisions, interventions, and treatments can be made for each group of patients. Similarly, stability guarantees are desired in designing personalized learning environments or marketing campaigns to ensure that no individual wants to deviate from their assigned cluster. Furthermore the focus on individual utility in IP stability (a clustering is only stable if every individual is "happy") enforces a notion of individual fairness in clustering.

In addition to its natural connections to cluster stability, algorithmic fairness, and Nash equilibria, IP stability is also algorithmically interesting in its own right. While clustering is well-studied with respect to global objective functions (e.g. the objectives of centroid-based clustering such as $k$-means

---

[*]Significant part of the work was done while P.S. was a Ph.D. candidate at Northwestern University.

37th Conference on Neural Information Processing Systems (NeurIPS 2023).

or correlation/hierarchical clustering), less is known when the goal is to partition the dataset such that every point in the dataset is individually satisfied with the solution. Thus, IP stability also serves as a natural and motivated model of individual preferences in clustering.

## 1.1 Problem Statement and Preliminaries

The main objective of our clustering algorithms is to achieve IP stability given a set $P$ of $n$ points lying in a metric space $(M, d)$ and $k$, the number of clusters.

**Definition 1.1** (Individual Preference (IP) Stability [1])**.** The goal is to find a disjoint $k$-clustering $\mathcal{C} = (C_1, \cdots, C_k)$ of $P$ such that every point, *on average*, is closer to the points of its own cluster than to the points in any other cluster. Formally, for all $v \in P$, let $C(v)$ denote the cluster that contains $v$. We say that $v \in P$ is IP stable with respect to $\mathcal{C}$ if either $C(v) = \{v\}$ or for every $C' \in \mathcal{C}$ with $C' \neq C$,

$$\frac{1}{|C(v)| - 1} \sum_{u \in C(v)} d(v, u) \leq \frac{1}{|C'|} \sum_{u \in C'} d(v, u). \tag{1}$$

The clustering $\mathcal{C}$ is 1-IP stable (or simply IP stable) if and only if every $v \in P$ is stable with respect to $\mathcal{C}$.

Ahmadi et al. [1] showed that an arbitrary dataset may not admit an IP stable clustering. This can be the case even when $n = 4$. Furthermore, they proved that it is NP-hard to decide whether a given a set of points have an IP stable $k$-clustering, even for $k = 2$. This naturally motivates the study of the relaxations of IP stability.

**Definition 1.2** (Approximate IP Stability)**.** A $k$-clustering $\mathcal{C} = (C_1, \cdots, C_k)$ of $P$ is $\alpha$-approximate IP stable, or simply $\alpha$-IP stable, if for every point $v \in P$, the following holds: either $C(v) = \{v\}$ or for every $C' \in \mathcal{C}$ and $C' \neq C$,

$$\frac{1}{|C(v)| - 1} \sum_{u \in C(v)} d(v, u) \leq \frac{\alpha}{|C'|} \sum_{u \in C'} d(v, u). \tag{2}$$

The work of [1] proposed algorithms to outputting IP stable clusterings on the one-dimensional line for any value of $k$ and on tree metrics for $k = 2$. The first result implies an $O(n)$-IP stable clustering for general metrics, by applying a standard $O(n)$-distortion embedding to one-dimensional Euclidean space. In addition, they give a bicriteria approximation that discards an $\varepsilon$-fraction of the input points and outputs a $O\left(\frac{\log^2 n}{\varepsilon}\right)$-IP stable clustering for the remaining points.

Given the prior results, it is natural to ask if the $O(n)$ factor for IP stable clustering given in [1] can be improved.

## 1.2 Our Results

**New Approximations.**   Improving on the $O(n)$-IP stable algorithm in [1], we present a deterministic algorithm which for general metrics obtains an $O(1)$-IP stable $k$-clustering, for any value of $k$. Note that given the existence of instances without 1-IP stable clusterings, our approximation factor is optimal up to a constant factor.

**Theorem 1.3.** *(Informal; see Theorem 3.1) Given a set $P$ of $n$ points in a metric space $(M, d)$ and a number of desired clusters $k \leq n$, there exists an algorithm that computes an $O(1)$-IP stable $k$-clustering of $P$ in polynomial time.*

Our algorithm outputs a clustering with an even stronger guarantee that we call uniform (approximate) IP stability. Specifically, for some global parameter $r$ and for every point $v \in P$, the average distance from $v$ to points in its own cluster is upper bounded by $O(r)$ and the average distance from $v$ to points in any other cluster is lower bounded by $\Omega(r)$. Note that the general condition of $O(1)$-IP stability would allow for a different value of $r$ for each $v$.

We again emphasize that Theorem 1.3 implies that an $O(1)$-IP stable clustering always exists, where prior to this work, only the $O(n)$ bound from [1] was known for general metrics.

**Additional $k$-center clustering guarantee.** The clustering outputted by our algorithm satisfies additional desirable properties beyond $O(1)$-IP stability. In the $k$-center problem, we are given $n$ points in a metric space, and our goal is to pick $k$ centers as to minimize the maximal distance of any point to the nearest center. The clustering outputted by our algorithm from Theorem 1.3 has the added benefit of being a constant factor approximation to the $k$-center problem in the sense that if the optimal $k$-center solution has value $r_0$, then the diameter of each cluster outputted by the algorithm is $O(r_0)$. In fact, we argue that IP stability is more meaningful when we also seek a solution that optimizes some clustering objective. If we only ask for IP stability, there are instances where it is easy to obtain $O(1)$-IP stable clusterings, but where such clusterings do not provide insightful information in a typical clustering application. Indeed, as we will show in Appendix B, randomly $k$-coloring the nodes of an unweighted, undirected graph (where the distance between two nodes is the number of edges on the shortest path between them), gives an $O(1)$-IP stable clustering when $k \leq O\left(\frac{\sqrt{n}}{\log n}\right)$. Our result on trees demonstrates the idiosyncrasies of individual objectives thus our work raises further interesting questions about studying standard global clustering objectives under the restriction that the solutions are also (approximately) IP stable.

**Max and Min-IP Stability.** Lastly, we introduce a notion of $f$-IP stability, generalizing IP stability.

**Definition 1.4** ($f$-IP Stability). Let $(M, d)$ be a metric space, $P$ a set of $n$ points of $M$, and $k$ the desired number of partitions. Let $f : P \times 2^P \to \mathbb{R}^{\geq 0}$ be a function which takes in a point $v \in P$, a subset $C$ of $P$, and outputs a non-negative real number. we say that a $k$-clustering $\mathcal{C} = (C_1, \cdots, C_k)$ of $P$ is $f$-IP stable if for every point $v \in P$, the following holds: either $C(v) = \{v\}$ or for every $C' \in \mathcal{C}$ and $C' \neq C$,

$$f\left(v, C(v) \setminus \{v\}\right) \leq f\left(v, C'\right). \tag{3}$$

Note that the standard setting of IP stability given in Definition 1.1 corresponds to the case where $f(v, C) = (1/|C|) \times \sum_{v' \in C} d(v, v')$. The formulation of $f$-IP stability, therefore, extends IP stability beyond average distances and allows for alternative objectives that may be more desirable in certain settings. For instance, in hierarchical clustering, average, minimum, and maximum distance measures are well-studied.

In particular, we focus on max-distance and min-distance in the definition of $f$-IP stable clustering in addition to average distance (which is just Definition 1.1), where $f(v, C) = \max_{v' \in C} d(v, v')$ and $f(v, C) = \min_{v' \in C} d(v, v')$. We show that in both the max and min distance formulations, we can solve the corresponding $f$-IP stable clustering (nearly) optimally in polynomial time. We provide the following result:

**Theorem 1.5** (Informal; see Theorem 4.1 and Theorem 4.2). *In any metric space, Min-IP stable clustering can be solved optimally and Max-IP stable clustering can be solved approximately within a factor of $3$, in polynomial time.*

We show that the standard greedy algorithm of $k$-center, a.k.a, the Gonzalez's algorithm [15], yields a 3-approximate Max-IP stable clustering. Moreover, we present a conceptually clean algorithm which is motivated by considering the minimum spanning tree (MST) to output a Min-IP stable clustering. This implies that unlike the average distance formulation of IP stable clustering, a Min-IP stable clustering always exists. Both algorithms work in general metrics.

| Metric | Approximation Factor | Reference | Remark |
|---|---|---|---|
| 1D Line metric | 1 | [1] | |
| Weighted tree | 1 | [1] | Only for $k = 2$ |
| General metric | $O(n)$ | [1] | |
| General metric | $O(1)$ | **This work** | |

Table 1: Our results on IP stable $k$-clustering of $n$ points. All algorithms run in polynomial time.

**Empirical Evaluations.** We experimentally evaluate our $O(1)$-IP stable clustering algorithm against $k$-means++, which is the empirically best-known algorithm in [1]. We also compare $k$-means++ with our optimal algorithm for Min-IP stability. We run experiments on the Adult data set[2]

---

[2]https://archive.ics.uci.edu/ml/datasets/adult; see [18].

used by [1]. For IP stability, we also use four more datasets from UCI ML repositoriy [11] and a synthetic data set designed to be a hard instance for $k$-means++. On the Adult data set, our algorithm performs slightly worse than $k$-means++ for IP stability. This is consistent with the empirical results of [1]. On the hard instance[3], our algorithm performs better than $k$-means++, demonstrating that the algorithm proposed in this paper is more robust than $k$-means++. Furthermore for Min-IP stability, we empirically demonstrate that $k$-means++ can have an approximation factors which are up to a factor of **5x** worse than our algorithm. We refer to Section 5 and Appendix C for more details.

### 1.3 Technical Overview

The main contribution is our $O(1)$-approximation algorithm for IP stable clustering for general metrics. We discuss the proof technique used to obtain this result. Our algorithm comprises two steps. We first show that for any radius $r$, we can find a clustering $\mathcal{C} = (C_1, \ldots, C_t)$ such that (a) each cluster has diameter $O(r)$, and (b) the average distance from a point in a cluster to the points of any other cluster is $\Omega(r)$.

Conditions (a) and (b) are achieved through a ball carving technique, where we iteratively pick centers $q_i$ of distance $> 6r$ to previous centers such that the radius $r$ ball $B(q_i, r)$ centered at $q_i$ contains a maximal number of points, say $s_i$. For each of these balls, we initialize a cluster $D_i$ containing the $s_i$ points of $B(q_i, r)$. We next consider the annulus $B(q_i, 3r) \setminus B(q_i, 2r)$. If this annulus contains less than $s_i$ points, we include all points from $B(q_i, 3r)$ in $D_i$. Otherwise, we include *any* $s_i$ points in $D_i$ from the annulus. We assign each unassigned point to the *first* center picked by our algorithm and is within distance $O(r)$ to the point. This is a subtle but crucial component of the algorithm as the more natural "assign to the closest center" approach fails to obtain $O(1)$-IP stability.

One issue remains. With this approach, we have no guarantee on the number of clusters. We solve this by merging some of these clusters while still maintaining that the final clusters have radius $O(r)$. This may not be possible for any choice of $r$. Thus the second step is to find the right choice of $r$. We first run the greedy algorithm of $k$-center and let $r_0$ be the minimal distance between centers we can run the ball carving algorithm $r = cr_0$ for a sufficiently small constant $c$. Then if we assign each cluster of $\mathcal{C}$ to its nearest $k$-center, we do indeed maintain the property that all clusters have diameter $O(r)$, and since $c$ is a small enough constant, all the clusters will be non-empty. The final number of clusters will therefore be $k$. As an added benefit of using the greedy algorithm for $k$-center as a subroutine, we obtain that the diameter of each cluster is also $O(r_0)$, namely the output clustering is a constant factor approximation to $k$-center.

### 1.4 Related Work

**Fair Clustering.** One of the main motivations of IP stable clustering is its interpretation as a notion of individual fairness for clustering [1]. Individual fairness was first introduced by [12] for the classification task, where, at high-level, the authors aim for a classifier that gives "similar predictions" for "similar" data points. Recently, other formulations of individual fairness have been studied for clustering [17, 2, 7, 8], too. [17] proposed a notion of fairness for centroid-based clustering: given a set of $n$ points $P$ and the number of clusters $k$, for each point, a center must be picked among its $(n/k)$-th closest neighbors. The optimization variant of it was later studied by [19, 20, 24].[7] studied a pairwise notion of fairness in which data points represent people who gain some benefit from being clustered together. In a subsequent work, [6] introduced a stochastic variant of this notion. [2] studied the setting in which the output is a distribution over centers and "similar" points are required to have "similar" centers distributions.

**Stability in Clustering.** Designing efficient clustering algorithms under notions of stability is a well-studied problem[4]. Among the various notion of stability, *average stability* is the most relevant to our model [4]. In particular, they showed that if there is a ground-truth clustering satisfying the requirement of Equation (1) with an additive gap of $\gamma > 0$, then it is possible to recover the solution in the list model where the list size is exponential in $1/\gamma$. Similar types of guarantees are shown in the work by [9]. While this line of research mainly focuses on presenting faster algorithms utilizing the

---

[3]The construction of this hard instance is available in the appendix of [1].
[4]For a comprehensive survey on this topic, refer to [3].

strong stability conditions, the focus of IP stable clustering is whether we can recover such stability properties in general instances, either exactly or approximately.

**Hedonic Games.** Another game-theoretic study of clustering is hedonic games [10, 5, 13]. In a hedonic game, players choose to form coalitions (i.e., clusters) based on their utility. Our work differs from theirs, since we do not model the data points as selfish players. In a related work, [23] proposes another utility measure for hedonic clustering games on graphs. In particular, they define a closeness utility, where the utility of node $i$ in cluster $C$ is the ratio between the number of nodes in $C$ adjacent to $i$ and the sum of distances from $i$ to other nodes in $C$. This measure is incomparable to IP stability. In addition, their work focuses only on clustering in graphs while we consider general metrics.

## 2 Preliminaries and Notations

We let $(M, d)$ denote a metric space, where $d$ is the underlying distance function. We let $P$ denote a fixed set of points of $M$. Here $P$ may contain multiple copies of the same point. For a given point $x \in P$ and radius $r \geq 0$, we denote by $B(x, r) = \{y \in P \mid d(x, y) \leq r\}$, the ball of radius $r$ centered at $x$. For two subsets $X, Y \subseteq P$, we denote by $d(X, Y) = \inf_{x \in X, y \in Y} d(x, y)$. Throughout the paper, $X$ and $Y$ will always be finite and then the infimum can be replaced by a minimum. For $x \in P$ and $Y \subseteq P$, we simply write $d(x, Y)$ for $d(\{x\}, Y)$. Finally, for $X \subseteq P$, we denote by $\mathrm{diam}(X) = \sup_{x, y \in X} d(x, y)$, the diameter of the set $X$. Again, $X$ will always be finite, so the supremum can be replaced by a maximum.

## 3 Constant-Factor IP Stable Clustering

In this section, we prove our main result: For a set $P = \{x_1, \ldots, x_n\}$ of $n$ points with a metric $d$ and every $k \leq n$, there exists a $k$-clustering $\mathcal{C} = (C_1, \ldots, C_k)$ of $P$ which is $O(1)$-approximate IP stable. Moreover, such a clustering can be found in time $\widetilde{O}(n^2 T)$, where $T$ is an upper bound on the time it takes to compute the distance between two points of $P$.

**Algorithm** Our algorithm uses a subroutine, Algorithm 1, which takes as input $P$ and a radius $r \in \mathbb{R}$ and returns a $t$-clustering $\mathcal{D} = (D_1, \ldots, D_t)$ of $P$ with the properties that (1) for any $1 \leq i \leq t$, the maximum distance between any two points of $D_i$ is $O(r)$, and (2) for any $x \in P$ and any $i$ such that $x \notin D_i$, the average distance from $x$ to points of $D_i$ is $\Omega(r)$. These two properties ensure that $\mathcal{D}$ is $O(1)$-approximate IP stable. However, we have no control on the number of clusters $t$ that the algorithm produces. To remedy this, we first run a greedy $k$-center algorithm on $P$ to obtain a set of centers $\{c_1, \ldots, c_k\}$ and let $r_0$ denote the maximum distance from a point of $P$ to the nearest center. We then run Algorithm 1 with input radius $r = cr_0$ for some small constant $c$. This gives a clustering $\mathcal{D} = (D_1, \ldots, D_t)$ where $t \geq k$. Moreover, we show that if we assign each cluster of $\mathcal{D}$ to the nearest center in $\{c_1, \ldots, c_k\}$ (in terms of the minimum distance from a point of the cluster to the center), we obtain a $k$-clustering $\mathcal{C} = (C_1, \ldots, C_k)$ which is $O(1)$-approximate IP stable. The combined algorithm is Algorithm 2.

We now describe the details of Algorithm 1. The algorithm takes as input $n$ points $x_1, \ldots, x_n$ of a metric space $(M, d)$ and a radius $r$. It first initializes a set $Q = \emptyset$ and then iteratively adds points $x$ from $P$ to $Q$ that are of distance greater than $6r$ from points already in $Q$ such that $|B(x, r)|$, the number of points of $P$ within radius $r$ of $x$, is maximized. This is line 5–6 of the algorithm. Whenever a point $q_i$ is added to $Q$, we define the annulus $A_i := B(q_i, 3r) \setminus B(q_i, 2r)$. We further let $s_i = |B(q_i, r)|$. At this point the algorithm splits into two cases. If $|A_i| \geq s_i$, we initialize a cluster $D_i$ which consists of the $s_i$ points in $B(x, r)$ and any arbitrarily chosen $s_i$ points in $A_i$. This is line 8–9 of the algorithm. If on the other hand $|A_i| < s$, we define $D_i := B(q_i, 3r)$, namely $D_i$ contains all points of $P$ within distance $3r$ from $q_i$. This is line 10 of the algorithm. After iteratively picking the points $q_i$ and initializing the clusters $D_i$, we assign the remaining points as follows. For any point $x \in P \setminus \bigcup_i D_i$, we find the minimum $i$ such that $d(x, q_i) \leq 7r$ and assign $x$ to $D_i$. This is line 13–16 of the algorithm. We finally return the clustering $\mathcal{D} = (D_1, \ldots, D_t)$.

We next describe the details of Algorithm 2. The algorithm iteratively pick $k$ centers $c_1, \ldots, c_k$ from $P$ for each center maximizing the minimum distance to previously chosen centers. For each center $c_i$, it initializes a cluster, starting with $C_i = \{c_i\}$. This is line 4–7 of the algorithm. Letting $r_0$ be

---

**Algorithm 1** BALL-CARVING

1: **Input**: A set $P = \{x_1, \ldots, x_n\}$ of $n$ points with a metric $d$ and a radius $r > 0$.
2: **Output**: Clustering $\mathcal{D} = (D_1, \ldots, D_t)$ of $P$.
3: $Q \leftarrow \emptyset$, $i \leftarrow 1$
4: **while** there exists $x \in P$ with $d(x, Q) > 6r$ **do**
5: $\quad q_i \leftarrow \arg\max_{x \in P: d(x,Q) > 6r} |B(x, r)|$
6: $\quad Q \leftarrow Q \cup \{q_i\}$, $s_i \leftarrow |B(q_i, r)|$, $A_i \leftarrow B(q_i, 3r) \setminus B(q_i, 2r)$
7: $\quad$ **if** $|A_i| \geq s_i$
8: $\quad\quad S_i \leftarrow$ any set of $s_i$ points from $A_i$
9: $\quad\quad D_i \leftarrow B(q_i, r) \cup S_i$
10: $\quad$ **else** $D_i \leftarrow B(q_i, 3r_i)$
11: $\quad i \leftarrow i + 1$
12: **end while**
13: **for** $x \in P$ assigned to no $D_i$ **do**
14: $\quad j \leftarrow \min\{i \mid d(x, q_i) \leq 7r\}$
15: $\quad D_j \leftarrow D_j \cup \{x\}$
16: **end for**
17: $t \leftarrow |Q|$
18: **return** $\mathcal{D} = (D_1, \ldots, D_t)$

---

**Algorithm 2** IP-CLUSTERING

1: **Input**: Set $P = \{x_1, \ldots, x_n\}$ of $n$ points with a metric $d$ and integer $k$ with $2 \leq k \leq n$.
2: **Output**: $k$-clustering $\mathcal{C} = (C_1, \ldots, C_k)$ of $P$.
3: $S \leftarrow \emptyset$
4: **for** $i = 1, \ldots, k$ **do**
5: $\quad c_i \leftarrow \arg\max_{x \in P}\{d(x, S)\}$
6: $\quad S \leftarrow S \cup \{c_i\}$, $C_i \leftarrow \{c_i\}$
7: **end for**
8: $r_0 \leftarrow \min\{d(c_i, c_j) \mid 1 \leq i < j \leq k\}$
9: $\mathcal{D} \leftarrow$ BALL-CARVING$(P, r_0/15)$
10: **for** $D \in \mathcal{D}$ **do**
11: $\quad j \leftarrow \arg\min_i\{d(c_i, D)\}$
12: $\quad C_j \leftarrow C_j \cup D$
13: **end for**
14: **return** $\mathcal{C} = (C_1, \ldots, C_k)$

---

the minimum distance between pairs of distinct centers, the algorithm runs Algorithm 1 on $P$ with input radius $r = r_0/15$ (line 8–9). This produces a clustering $\mathcal{D}$. In the final step, we iterate over the clusters $D$ of $\mathcal{D}$, assigning $D$ to the $C_i$ for which $d(c_i, D)$ is minimized (line 11–13). We finally return the clustering $(C_1, \ldots, C_k)$.

**Analysis** We now analyze our algorithm and provide its main guarantees.

**Theorem 3.1.** *Algorithm 2 returns an $O(1)$-approximate IP stable $k$ clustering in time $O(n^2 T + n^2 \log n)$. Furthermore, the solution is also a constant factor approximation to the $k$-center problem.*

In order to prove this theorem, we require the following lemma on Algorithm 1.

**Lemma 3.2.** *Let $(D_1, \ldots, D_t)$ be the clustering output by Algorithm 1. For each $i \in [t]$, the diameter of $D_i$ is at most $14r$. Further, for $x \in D_i$ and $j \neq i$, the average distance from $x$ to points of $D_j$ is at least $\frac{r}{4}$.*

Given Lemma 3.2, we can prove the the main result.

*Proof of Theorem 3.1.* We first argue correctness. As each $c_i$ was chosen to maximize the minimal distance to points $c_j$ already in $S$, for any $x \in P$, it holds that $\min\{d(x, c_i) \mid i \in [k]\} \leq r_0$. By Lemma 3.2, in the clustering $\mathcal{D}$ output by BALL-CARVING$(P, r_0/15)$ each cluster has diameter at most $\frac{14}{15}r_0 < r_0$, and thus, for each $i \in [k]$, the cluster $D \in \mathcal{D}$ which contains $c_i$ will be included in

$C_i$ in the final clustering. Indeed, in line 11 of Algorithm 2, $d(c_i, D) = 0$ whereas $d(c_j, D) \geq \frac{1}{15}r_0$ for all $j \neq i$. Thus, each cluster in $(C_1, \ldots, C_k)$ is non-empty. Secondly, the diameter of each cluster is at most $4r_0$, namely, for each two points $x, x' \in C_i$, they are both within distance $r_0 + \frac{14}{15}r_0 < 2r_0$ of $c_i$. Finally, by Lemma 3.2, for $x \in D_i$ and $j \neq i$, the average distance from $x$ to points of $D_j$ is at least $\frac{r_0}{60}$. Since, $\mathcal{C}$ is a coarsening of $\mathcal{D}$, i.e., each cluster of $\mathcal{C}$ is the disjoint union of some of the clusters in $\mathcal{D}$, it is straightforward to check that the same property holds for the clustering $\mathcal{C}$. Thus $\mathcal{C}$ is $O(1)$-approximate IP stable.

We now analyze the running time. We claim that Algorithm 2 can be implemented to run in $O(n^2 T + n^2 \log n)$ time, where $T$ is the time to compute the distance between any two points in the metric space. First, we can query all pairs to form the $n \times n$ distance matrix $A$. Then we sort $A$ along every row to form the matrix $A'$. Given $A$ and $A'$, we easily implement our algorithms as follows.

First, we argue about the greedy $k$-center steps of Algorithm 2, namely, the for loop on line 4. The most straightforward implementation computes the distance from every point to new chosen centers. At the end, we have computed at most $nk$ distances from points to centers which can be looked up in $A$ in time $O(nk) = O(n^2)$ as $k \leq n$. In line 8, we only look at every entry of $A$ at most once so the total time is also $O(n^2)$. The same reasoning also holds for the for loop on line 10. It remains to analyze the runtime.

Given $r$, Alg. 1 can be implemented as follows. First, we calculate the size of $|B(x, r)|$ for every point $x$ in our dataset. This can easily be done by binary searching on the value of $r$ along each of the (sorted) rows of $A'$, which takes $O(n \log n)$ time in total. We can similarly calculate the sizes of $|B(x, 2r)|$ and $|B(x, 3r)|$, and thus the number of points in the annulus $|B(x, 3r) \setminus B(x, 2r)|$ in the same time to initialize the clusters $D_i$. Similar to the $k$-center reasoning above, we can also pick the centers in Algorithm 1 which are $> 6r$ apart iteratively by just calculating the distances from points to the chosen centers so far. This costs at most $O(n^2)$ time, since there are at most $n$ centers. After initializing the clusters $D_i$, we finally need to assign the remaining unassigned points (line 13–16). This can easily be done in time $O(n)$ per point, namely for each unassigned point $x$, we calculate its distance to each $q_i$ assigning it to $D_i$ where $i$ is minimal such that $d(x, q_i) \leq 7r$. The total time for this is then $O(n^2)$. The $k$-center guarantees follow from our choice of $r_0$ and Lemma 3.2. $\square$

*Remark* 3.3. We note that the runtime can possibly be improved if we assume special structure about the metric space (e.g., Euclidean metric). See Appendix A for a discussion.

We now prove Lemma 3.2.

*Proof of Lemma 3.2.* The upper bound on the diameter of each cluster follows from the fact that for any cluster $D_i$ in the final clustering $\mathcal{D} = \{D_1, \ldots, D_t\}$, and any $x \in D_i$, it holds that $d(x, q_i) \leq 7r$. The main challenge is to prove the lower bound on the average distance from $x \in D_i$ to $D_j$ where $j \neq i$.

Suppose for contradiction that, there exists $i, j$ with $i \neq j$ and $x \in D_i$ such that the average distance from $x$ to $D_j$ is smaller than $r/4$, i.e., $\frac{1}{|D_j|} \sum_{y \in D_j} d(x, y) < r/4$. Then, it in particular holds that $|B(x, r/2) \cap D_j| > |D_j|/2$, namely the ball of radius $r/2$ centered at $x$ contains more than half the points of $D_j$. We split the analysis into two cases corresponding to the if-else statements in line 7–10 of the algorithm.

**Case 1: $|A_j| \geq s_j$:** In this case, cluster $D_j$ consists of at least $2s_j$ points, namely the $s_j$ points in $B(q_j, r)$ and the set $S_j$ of $s_j$ points in $A_j$ assigned to $D_j$ in line 8–9 of the algorithm. It follows from the preceding paragraph that, $|B(x, r/2) \cap D_j| > s_j$. Now, when $q_j$ was added to $Q$, it was chosen as to maximize the number of points in $B(q_j, r)$ under the constraint that $q_j$ had distance greater than $6r$ to previously chosen points of $Q$. Since $|B(x, r)| > |B(x, r/2)| > |B(q_j, r)|$, at the point where $q_j$ was chosen, $Q$ already contained some point $q_{j_0}$ (with $j_0 < j$) of distance at most $6r$ to $x$ and thus of distance at most $7r$ to any point of $B(x, r/2)$. It follows that $B(x, r/2) \cap D_j$ contains no point assigned during line 13–16 of the algorithm. Indeed, by the assignment rule, such a point $y$ would have been assigned to either $D_{j_0}$ or potentially an even earlier initialized cluster of distance at most $7r$ to $y$. Thus, $B(x, r/2) \cap D_j$ is contained in the set $B(q_j, r) \cup S_j$. However, $|B(q_j, r)| = |S_j| = s_j$ and moreover, for $(y_1, y_2) \in B(q_j, r) \times S_j$, it holds that $d(y_1, y_2) > r$. In particular, no ball of radius $r/2$ can contain more than $s_j$ points of $B(q_j, r) \cup S_j$. As $|B(x, r/2) \cap D_j| > s_j$, this is a contradiction.

**Case 2:** $|A_j| < s_j$**:**  In this case, $D_j$ includes all points in $B(q_j, 3r)$. As $x \notin D_j$, we must have that $x \notin B(q_j, 3r)$ and in particular, the ball $B(x, r/2)$ does not intersect $B(q_j, r)$. Thus,

$$|D_j| \geq |B(x, r/2) \cap D_j| + |B(q_j, r) \cap D_j| > |D_j|/2 + s_j,$$

so $|D_j| > 2s_j$, and finally, $|B(x, r/2) \cap D_j| > |D_j|/2 > s_j$. Similarly to case 1, $B(x, r/2) \cap D_j$ contains no points assigned during line 13– 16 of the algorithm. Moreover, $B(x, r/2) \cap B(q_j, 3r) \subseteq A_j$. In particular, $B(x, r/2) \cap D_j \subseteq S_j$, a contradiction as $|S_j| = s_j$ but $|B(x, r/2) \cap D_j| > s_j$. $\qquad \square$

## 4   Min and Max-IP Stable Clustering

The Min-IP stable clustering aims to ensure that for any point $x$, the *minimum* distance to a point in the cluster of $x$ is at most the minimum distance to a point in any other cluster. We show that a Min-IP stable $k$-clustering always exists for any value of $k \in [n]$ and moreover, can be found by a simple algorithm (Algorithm 3).

---

**Algorithm 3** MIN-IP-CLUSTERING

1: **Input**: Pointset $P = \{x_1, \ldots, x_n\}$ from a metric space $(M, d)$ and integer $k$ with $2 \leq k \leq n$.
2: **Output**: $k$-clustering $\mathcal{C} = (C_1, \ldots, C_k)$ of $P$.
3: $L \leftarrow \{(x_i, x_j)\}_{1 \leq i < j \leq n}$ sorted according to $d(x_i, x_j)$
4: $E \leftarrow \emptyset$
5: **while** $G = (P, E)$ has $> k$ connected components **do**
6:     $e \leftarrow$ an edge $e = (x, y)$ in $L$ with $d(x, y)$ minimal.
7:     $L \leftarrow L \setminus \{e\}$
8:     **if** $e$ connects different connected components of $G$ **then** $E \leftarrow E \cup \{e\}$
9: **end while**
10: **return** the connected components $(C_1, \ldots, C_k)$ of $G$.

---

The algorithm is identical to Kruskal's algorithm for finding a minimum spanning tree except that it stops as soon as it has constructed a forest with $k$ connected components. First, it initializes a graph $G = (V, E)$ with $V = P$ and $E = \emptyset$. Next, it computes all distances $d(x_i, x_j)$ between pairs of points $(x_i, x_j)$ of $P$ and sorts the pairs $(x_i, x_j)$ according to these distances. Finally, it goes through this sorted list adding each edge $(x_i, x_j)$ to $E$ if it connects different connected components of $G$. After computing the distances, it is well known that this algorithm can be made to run in time $O(n^2 \log n)$, so the total running time is $O(n^2(T + \log n))$ where $T$ is the time to compute the distance between a single pair of points.

**Theorem 4.1.** *The $k$-clustering output by Algorithm 3 is a Min-IP stable clustering.*

*Proof.* Let $\mathcal{C}$ be the clustering output by the algorithm. Conditions (1) and (2) in the definition of a min-stable clustering are trivially satisfied. To prove that (3) holds, let $C \in \mathcal{C}$ with $|C| \geq 2$ and $x \in C$. Let $y_0 \neq x$ be a point in $C$ such that $(x, y_0) \in E$ (such an edge exists because $C$ is the connected component of $G$ containing $x$) and let $y_1$ be the closest point to $x$ in $P \setminus C$. When the algorithm added $(x, y_0)$ to $E$, $(x, y_1)$ was also a candidate choice of an edge between connected components of $G$. Since the algorithm chose the edge of minimal length with this property, $d(x, y_0) \leq d(x, y_1)$. Thus, we get the desired bound:

$$\min_{y \in C \setminus \{x\}} d(x, y) \leq d(x, y_0) \leq d(x, y_1) = \min_{y \in P \setminus C} d(x, y). \qquad \square$$

**Theorem 4.2.** *The solution output by the greedy algorithm of $k$-center is a 3-approximate Max-IP stable clustering.*

*Proof.* To recall, the greedy algorithm of $k$-center (aka Gonzalez algorithm [15]) starts with an arbitrary point as the first center and then goes through $k - 1$ iterations. In each iteration, it picks a new point as a center which is furthest from all previously picked centers. Let $c_1, \cdots, c_k$ denote the selected centers and let $r := \max_{v \in P} d(v, \{c_1, \cdots, c_k\})$. Then, each point is assigned to the cluster of its closest center. We denote the constructed clusters as $C_1, \cdots, C_k$. Now, for every $i \neq j \in [k]$ and each point $v \in C_i$, we consider two cases:

- $d(v, c_i) \le r/2$.

$$\max_{u_i \in C_i} d(v, u_i) \le d(v, c_i) + d(u_i, c_i) \le 3r/2,$$
$$\max_{u_j \in C_j} d(v, u_j) \ge d(v, c_j) \ge d(c_i, c_j) - d(v, c_i). \ge r/2$$

- $d(v, c_i) > r/2$.

$$\max_{u_i \in C_i} d(v, u_i) \le d(v, c_i) + d(u_i, c_i) \le 3d(v, c_i),$$
$$\max_{u_j \in C_j} d(v, u_j) \ge d(v, c_j) \ge d(v, c_i).$$

In both cases, $\max_{u_i \in C_i} d(v, u_i) \le 3 \max_{u_j \in C_j} d(v, u_j)$. $\square$

## 5  Experiments

While the goal and the main contributions of our paper are mainly theoretical, we also implement our optimal Min-IP clustering algorithm as well as extend the experimental results for IP stable clustering given in [1]. Our experiments demonstrate that our optimal Min-IP stable clustering algorithm is superior to $k$-means++, the strongest baseline in [1], and show that our IP clustering algorithm for average distances is practical on real world datasets and is competitive to $k$-means++ (which fails to find good stable clusterings in the worst case [1]). We give our experimental results for Min-IP stability and defer the rest of the empirical evaluations to Section C. All experiments were performed in Python 3. The results shown below are an average of 10 runs for $k$-means++.

**Metrics**   We measure the quality of a clustering using the same metrics used in [1] for standardization. Considering the question of $f$-IP stability (Definition 1.4), let the violation of a point $x$ be defined as $\mathrm{Vi}(x) = \max_{C_i \ne C(x)} \frac{f(x, C(x) \setminus \{x\})}{f(x, C_i)}$.

For example, setting $f(x, C) = \sum_{y \in C} d(x, y)/|C|$ corresponds to the standard IP stability objective and $f(x, C) = \min_{y \in C} d(x, y)$ is the Min-IP formulation. Note point $x$ is stable iff $\mathrm{Vi}(x) \le 1$.

We measure the extent to which a $k$-clustering $\mathcal{C} = (C_1, \ldots, C_k)$ of $P$ is (un-)stable by computing $\mathrm{MaxVi} = \max_{x \in P} \mathrm{Vi}(x)$ (i.e. maximum violation) and $\mathrm{MeanVi} = \sum_{x \in P} \mathrm{Vi}(x)/|P|$ (mean violation).

**Results**   For Min-IP stability, we have an optimal algorithm; it always return a stable clustering for all $k$. We see in Figures 1 that for the max and mean violation metrics, our algorithm outperforms $k$-means++ by up to a factor of **5x**, consistently across various values of $k$. $k$-means ++ can return a much worse clustering under Min-IP stability on real data, motivating the use of our theoretically-optimal algorithm in practice.

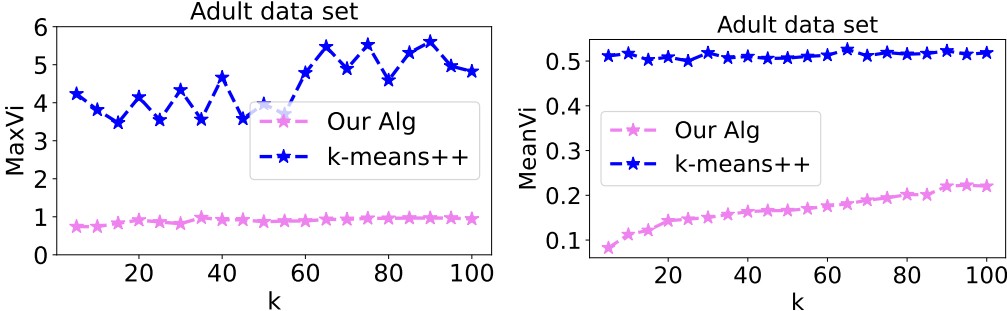

Figure 1: Maximum and mean violation for Min-IP stability for the Adult dataset, as used in [1]; lower values are better.

## 6    Conclusion

We presented a deterministic polynomial time algorithm which provides an $O(1)$-approximate IP stable clustering of $n$ points in a general metric space, improving on prior works which only guaranteed an $O(n)$-approximate IP stable clustering. We also generalized IP stability to $f$-stability and provided an algorithm which finds an exact Min-IP stable clustering and a 3-approximation for Max-IP stability, both of which hold for all $k$ and in general metric spaces.

## Acknowledgements

Anders Aamand is supported by DFF-International Postdoc Grant 0164-00022B from the Independent Research Fund Denmark and a Simons Investigator Award. Justin Chen and Allen Liu are supported in part by an NSF Graduate Research Fellowship under Grant No. 174530. Allen Liu is supported in part by a Hertz Fellowship. Fred Zhang is supported by ONR grant N00014-18-1-2562.

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

## A  Discussion on the Runtime of Algorithm 2

We remark that the runtime of our $O(1)$-approximate IP-stable clustering algorithm can potentially be improved if we assume special structure about the metric space, such as a tree or Euclidean metric. In special cases, we can improve the running time by appealing to particular properties of the metric which allow us to either calculate distances or implement our subroutines faster. For example for tree metrics, all distances can be calculated in $O(n^2)$ time, even though $T = O(n)$. Likewise for the Euclidean case, we can utilize specialized algorithms for computing the all pairs distance matrix, which obtain speedups over the naive methods [16], or use geometric point location data structures to quickly compute quantities such as $|B(x, r)|$ [22]. Our presentation is optimized for simplicity and generality so detailed discussions of specific metric spaces are beyond the scope of the work.

## B  Random Clustering in Unweighted Graphs

In this appendix, we show that for unweigthed, undirected, graphs (where the distance $d(u, v)$ between two vertices $u$ and $v$ is the length of the shortest path between them), randomly $k$-coloring the nodes gives an $O(1)$-approximate IP-stable clustering whenever $k = O(n^{1/2}/\log n)$.

We start with the following lemma.

**Lemma B.1.** *Let $\gamma = O(1)$ be a constant. There exists a constant $c > 0$ (depending on $\gamma$) such that the following holds: Let $T = (V, E)$ be an unweighted tree on $n$ nodes rooted at vertex $r$. Suppose that we randomly $k$-color the nodes of $T$. Let $V_i \subseteq V$ be the nodes of color $i$, let $X_i = \sum_{v \in V_i} d(r, v)$, and let $X = \sum_{v \in V} d(r, v)$. If $k \leq c\frac{\sqrt{n}}{\log n}$, then with probability $1 - O(n^{-\gamma})$, it holds that $X/2 \leq kX_i \leq 2X$ for all $i \in [k]$.*

*Proof.* We will fix $i$, and prove that the bound $X/2 \leq X_i \leq 2X$ holds with probability $1 - O(n^{-\gamma-1})$. Union bounding over all $i$ then gives the desired result. Let $\Delta = \max_{v \in V} d(r, v)$ be the maximum distance from the root to any vertex of the tree. We may assume that $\Delta \geq 5$ as otherwise the result follows directly from a simple Chernoff bound. Since the tree is unweighted and there exists a node $v$ of distance $\Delta$ to $r$, there must also exist nodes of distances $1, 2, \dots, \Delta - 1$ to $r$, namely the nodes on the path from $r$ to $v$. For the remaining nodes, we know that the distance is at least 1. Therefore,

$$\sum_{v \in V} d(r, v) \geq (n - \Delta - 1) + \sum_{j=1}^{\Delta} j = n + \binom{\Delta}{2} - 1 \geq n + \frac{\Delta^2}{3},$$

and so $\mu_i = \mathbb{E}[X_i] \geq \frac{n + \Delta^2/3}{k}$. Since the variables $(d(r, v)[v \in V_i])_{v \in V}$ sum to $X_i$ and are independent and bounded by $\Delta$, it follows by a Chernoff bound that for any $0 \leq \delta \leq 1$,

$$\Pr[|X_i - \mu_i| \geq \delta\mu_i] \leq 2\exp\left(-\frac{\delta^2 \mu_i}{3\Delta}\right).$$

By the AM-GM inequality,

$$\frac{\mu_i}{\Delta} \geq \frac{1}{k}\left(\frac{n}{\Delta} + \frac{\Delta}{3}\right) \geq \frac{2\sqrt{n}}{\sqrt{3}k}.$$

Putting $\delta = 1/2$, the bound above thus becomes

$$\Pr[|X_i - \mu_i| \geq \frac{\mu_i}{3}] \leq 2\exp\left(-\frac{\sqrt{n}}{6\sqrt{3}k}\right)$$

$$\leq 2\exp\left(-\frac{\sqrt{n}}{11k}\right) \leq 2n^{-\frac{1}{11c}},$$

where the last bound uses the assumption on the magnitude of $k$ in the lemma. Choosing $c = \frac{1}{11(\gamma+1)}$, the desired result follows. □

Next, we state our result on the $O(1)$-approximate IP-stability for randomly colored graphs.

**Theorem B.2.** *Let $\gamma = O(1)$ and $k \le c \frac{\sqrt{n}}{\log n}$ for a sufficiently small constant c. Let $G = (V, E)$ be an unweighted, undirected graph on $n$ nodes, and suppose that we k-color the vertices of G randomly. Let $V_i$ denote the nodes of color i. With probability at least $1 - n^{-\gamma}$, $(V_1, \ldots, V_k)$ forms an $O(1)$-approximate IP-clustering.*

*Proof.* Consider a node $u$ and let $X_i = \sum_{v \in V_i \setminus \{u\}} d(u, v)$. Node that the distances $d(u, v)$ are exactly the distances in a breath first search tree rooted at $v$. Thus, by Lemma B.1, the $X_i$'s are all within a constant factor of each other with probability $1 - O(n^{-\gamma-1})$. Moreover, a simple Chernoff bound shows that with the same high probability, $|V_i| = \frac{n}{k} + O\left(\sqrt{\frac{n \log n}{k}}\right) = \Theta\left(\frac{n}{k}\right)$ for all $i \in [k]$. In particular, the values $Y_i = \frac{X_i}{|V_i \setminus \{u\}|}$ for $i \in [k]$ also all lie within a constant factor of each other which implies that $u$ is $O(1)$-stable in the clustering $(V_1, \ldots, V_k)$. Union bounding over all nodes $u$, we find that with probability $1 - O(n^{-\gamma})$, $(V_1, \ldots, V_k)$ is an $O(1)$-approximate IP-clustering.  $\square$

*Remark* B.3. The assumed upper bound on $k$ in Theorem B.2 is necessary (even in terms of $\log n$). Indeed, consider a tree $T$ which is a star $S$ on $n - k \log k$ vertices along with a path $P$ of length $k \log k$ having one endpoint at the center $v$ of the star. With probability $\Omega(1)$, some color does not appear on $P$. We refer to this color as color 1. Now consider the color of the star center. With probability at least $9/10$, say, this color is different from 1 and appears $\Omega(\log k)$ times on $P$ with average distance $\Omega(k \log k)$ to the star center $v$. Let the star center have color 2. With high probability, each color appears $\Theta(n/k)$ times in $S$. Combining these bounds, we find that with constant probability, the average distance from $v$ to vertices of color 1 is $O(1)$, whereas the average distance from $v$ to vertices of color 2 is $\Omega\left(1 + \frac{k^2 (\log k)^2}{n}\right)$. In particular for the algorithm to give an $O(1)$-approximate IP-stable clustering, we need to assume that $k = O\left(\frac{\sqrt{n}}{\log n}\right)$.

## C   Additional Empirical Evaluations

We implement our $O(1)$-approximation algorithm for IP-clustering. These experiments extend those of [1] and confirm their experimental findings: $k$-means++ is a strong baseline for IP-stable clustering. Nevertheless, our algorithm is competitive with it while guaranteeing robustness against worst-case datasets, a property which $k$-means++ does not posses.

Our datasets are the following. There are three datasets from [11] used in [1], namely, `Adult`, `Drug` [14], and `IndianLiver`. We also add two additional datasets from UCI Machine Learning Repository [11], namely, `BreastCancer` and `Car`. For IP-clustering, we also consider a synthetic dataset which is the hard instance for $k$-means++ given in [1].

Our goal is to show that our IP-clustering algorithm is practical and in real world datasets, is competitive with respect to $k$-means++, which was the best algorithm in the experiments in [1]. Furthermore, our algorithm is robust and outperform $k$-means++ for worst case datasets.

As before, all experiments were performed in Python 3. We use the $k$-means++ implementation of Scikit-learn package [21]. We note that in the default implementation in Scikit-learn, $k$-means++ is initiated many times with different centroid seeds. The output is the best of 10 runs by default. As we want to have control of this behavior, we set the parameter `n_init=1` and then compute the average of many different runs.

Additionally to the metrics used in the main experimental section, we also compute the number of unstable points, defined as the size of the set $U = \{x \in M : x \text{ is not stable}\}$. In terms of clustering qualities, we additionally measure three quantities. First, we measure "cost", which is the average within-cluster distances. Formally, $Cost = \sum_{i=1}^{k} \frac{1}{\binom{|C_i|}{2}} \sum_{x,y \in C_i, x \neq y} d(x, y)$. We then measure $k$-center costs, defined as the maximum distances from any point to its center. Here, centers are given naturally from k-means++ and our algorithm. Finally, $k$-means costs, defined as k-means-cost $= \sum_{i=1}^{k} \frac{1}{|C_i|} \sum_{x,y \in C_i, x \neq y} d(x, y)^2$.

### C.1  Hard Instance for $k$-means++ for IP-Stability

We briefly describe the hard instance for $k$-means++ for the standard IP-stability formulation given in [1]; see their paper for full details. The hard instance consists of a gadget of size 4. In the seed-finding phase of $k$-means++, if it incorrectly picks two centers in the gadget, then the final clustering is not $\beta$-approximate IP-stable, where $\beta$ is a configurable parameter. The instance for $k$-clustering is produced by concatenating these gadgets together. In such an instance, with a constant probability, the clustering returned by $k$-means++ is not $\beta$-approximate IP-stable and in particular. We remark that the proof of Theorem 2 in [1] easily implies that $k$-means++ cannot have an approximation factor better than $n^c$ for some absolute constant $c > 0$, i.e., we can insure $\beta = \Omega(n^c)$. Here, we test both our algorithm and $k$-means++ in an instance with 8,000 points (for $k = 2,000$ clusters).

**IP-Stability results**  We first discuss five real dataset. We tested the algorithms for the range of $k$ up to 25. The result in Figures 2 and 3 is consistent with the experiments in the previous paper as we see that $k$-means++ is a very competitive algorithm for these datasets. For small number of clusters, our algorithm sometimes outperforms $k$-means++. We hypothesize that on these datasets, especially for large $k$, clusters which have low $k$-means cost separate the points well and therefore are good clusters for IP-stability.

Next we discuss the $k$-means++ hard instance. The instance used in Figure 3 was constructed with $\beta = 50$. We vary $k$ but omit the results for higher $k$ values since the outputs from both algorithms are stable. We remark that the empirical results with different $\beta$ gave qualitatively similar results. For maximum and mean violation, our algorithm outperforms $k$-means++ (Figure 3).

## D  Future Directions

There are multiple natural open questions following our work.

- Note that in some cases, $\alpha$-IP stable clusterings for $\alpha < 1$ may exist. On the other hand, in the hard example on $n = 4$ from [1], we know that there some constant $C > 1$ such that no $C$-IP stable clutering exists. For a given input, let $\alpha^*$ be the minimum value such that a $\alpha^*$-IP stable clustering exists. Is there an efficient algorithm which returns an $O(\alpha^*)$-IP stable clustering? Note that our algorithm satisfies this for $\alpha = \Omega(1)$. An even stronger result would be to find a PTAS which returns a $(1 + \varepsilon)\alpha^*$-IP stable clustering.

- For what specific metrics (other than the line or tree metrics with $k = 2$) can we get 1-IP stable clusterings efficiently?

- In addition to stability, it is desirable that a clustering algorithm also achieves strong global welfare guarantee. Our algorithm gives constant approximation for $k$-center. What about other metrics, such as $k$-means?

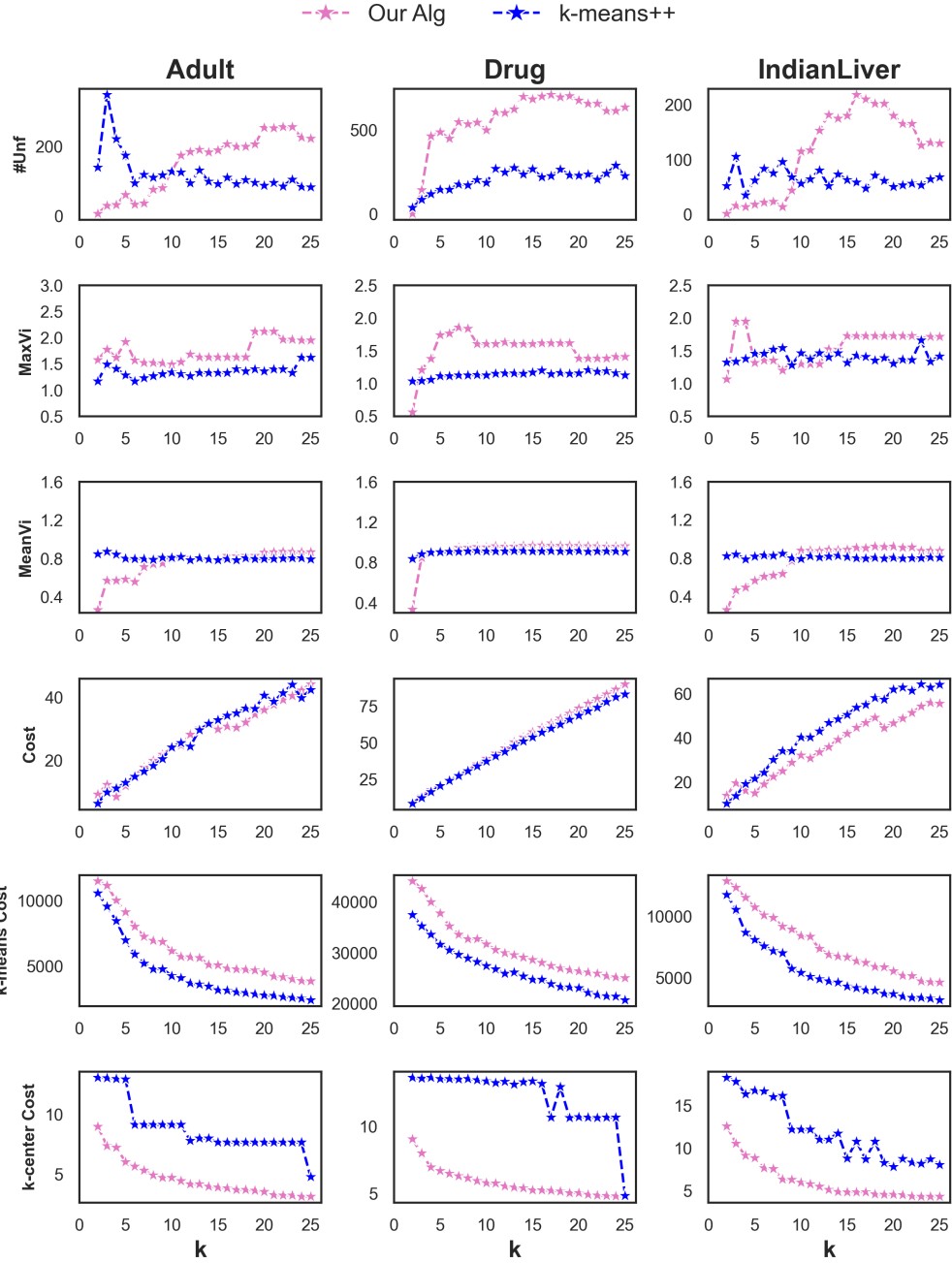

Figure 2: Experiment results on three datasets used in [1].

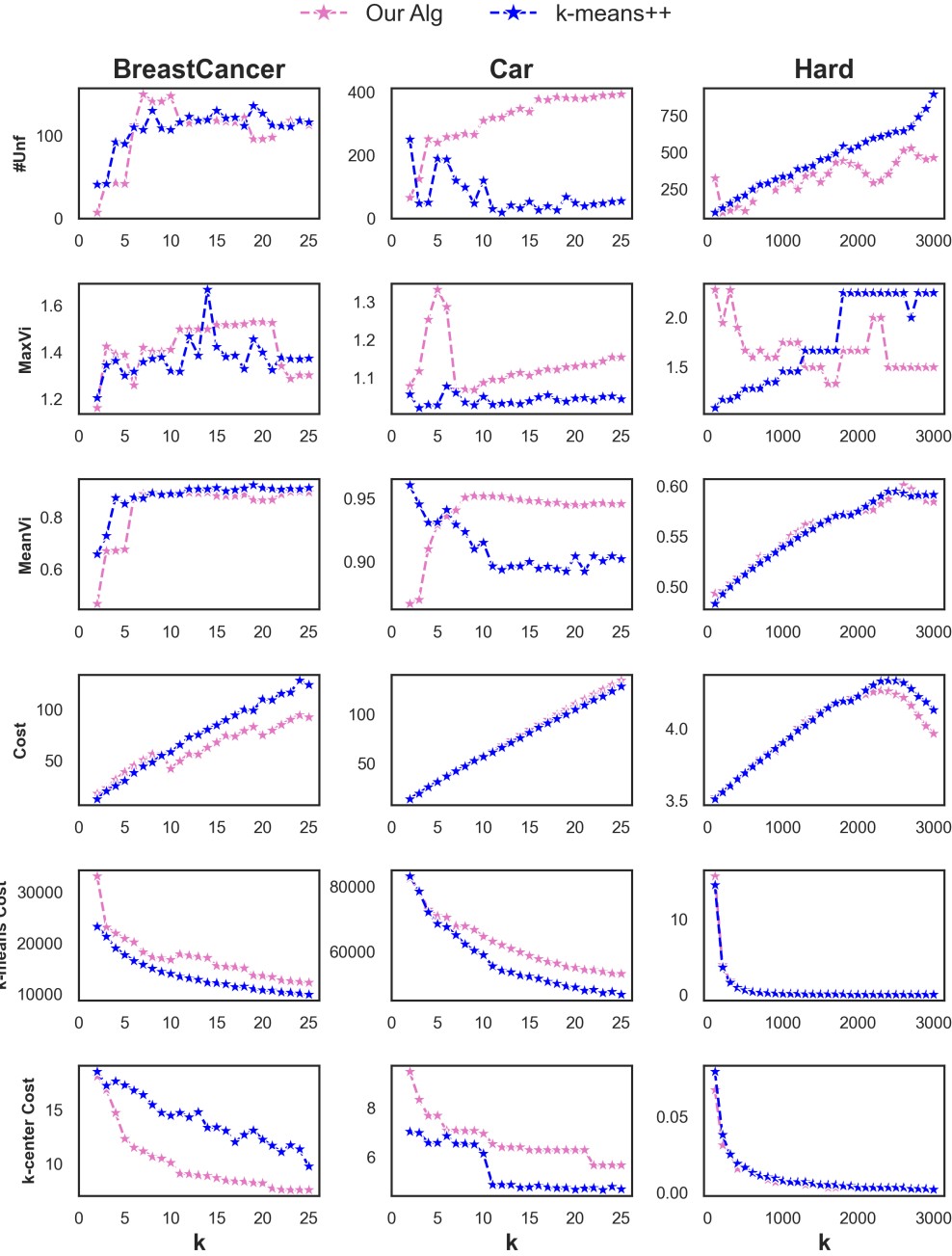

Figure 3: Additional experiment results on two real datasets and the synthetic dataset.

