# OpenReview forum: "Constant Approximation for Individual Preference Stable Clustering"
_NeurIPS.cc/2023/Conference — NeurIPS 2023 spotlight_

### Official Review · Reviewer_kydQ · 2023-07-05

**Soundness:** 4 excellent
**Presentation:** 3 good
**Contribution:** 3 good
**Rating:** 7
**Confidence:** 3

**Summary:**

This paper continues the study of individual preference stability (IP stability) initiated by Ahmadi et. al. (2022). Given a set of points in a metric space, a k-clustering is said to be 1-IP stable if for every point in the set, its average distance to the points in its cluster is at most its average distance to the points in a different cluster; so, in this sense, a point prefers its own cluster to other clusters. There are datasets for which a 1-IP stable clustering does not exist, and further, it is NP-hard to decide whether a dataset admits a 1-IP stable k-clustering. In light of this, it is natural to broaden the definition to approximate IP stability: a k-clustering is \alpha-IP stable if the average distance of a point to another cluster is at most \alpha times its average distance to its own cluster. The work of Ahmadi et. al. shows that an O(n)-IP stable clustering always exists and gives an algorithm for computing one. The present work closes the gap between the known lower and upper bounds: the authors propose an algorithm that always outputs an O(1)-stable clustering (so they also prove that one always exists, a fact not already established). Interestingly, the output clustering has the additional property that it is a constant factor approximation for k-center. Indeed, the greedy k-center algorithm of Gonzalez is a phase in their algorithm, and the "ball-carving" phase of their algorithm is similar in spirit to another approximation algorithm for k-center (although there are important nuances in the algorithm in the present work that differentiate it). The fact that the algorithm doubles as an approximation algorithm for k-center, the authors note, ensures that the O(1)-IP stable clustering is in some sense meaningful; for, they demonstrate that a random k-coloring a graph will produce an O(1)-IP stable clustering (for certain k), but of course such a clustering is not meaningful in general.

The authors also introduce two variants of IP stability. For Min-IP stability, in which "average distance" is replaced with "min distance" in the definition of IP stability, they show that the Single Link algorithm produces an exactly stable (optimal) clustering. For Max-IP stability (defined analogously), they show the greedy k-center algorithm of Gonzalez gives a 3-IP stable clustering.

Finally, the authors show experiments comparing their algorithms to k-means ++ as a baseline. While their algorithm performs slightly worse than k-means ++ for IP stability in general, they claim its robustness by showing a hard instance for which it outperforms k-means ++. They also show that in practice k-means ++ can be up to five times worse than their (optimal) algorithm for Min-IP stability.

**Strengths:**

- This work closes the (previously large) gap between upper and lower bounds for IP-stability. It demonstrates the existence of constant-stable clusterings and how to find them efficiently. It also provides a convincing argument that a clustering produced by the algorithm is meaningful by showing that it also gives a constant approximation for k-center, which I think is a key contribution given that the authors show, on the flip side, that a clustering can satisfy O(1)-stability but otherwise not uncover meaningful structure.

- The algorithm takes inspiration from approximation algorithms for k-center but there are important subtleties in the adaptation of these algorithms. The Ball-Carving algorithm (Algorithm 1) is a more nuanced version of an approximation algorithm for k-center (in which one guesses the optimal radius, and then repeatedly cuts out balls of that radius until no points remain). In Algorithm 1, more care is taken as to how to choose the center of the ball at each iteration (instead of being chosen arbitrarily as in the approximation algorithm)  as well as how to assign points to centers (instead of just carving out a ball of a certain radius). Finally, the centers from running the algorithm of Gonzalez are used to prune the number of clusters produced in the carving algorithm.

- The authors repurpose existing algorithms (Single Link clustering and the greedy algorithm of Gonzalez) to give guarantees for Min- and Max- IP stability. This is interesting in its own right because it shows that the notion of stability is intimately related to other clustering objectives.

**Weaknesses:**

- While Algorithms 1 and 2 are described very clearly, little motivation or intuition is given. For instance, it would be useful to provide examples that show the necessity of the nuances in the Ball-Carving algorithm.

- A discussion of how Algorithms 1/2 differ from or build on the algorithms of Ahmadi et. al. would help highlight the novelty of the contributions.

- A more specific comment: The paragraph on lines 66-70 seems inconsistent with later claims (unless I have misinterpreted something). It seems that if the parameter r is chosen to be anything, then the properties claimed in lines 66-70 hold, but potentially with more than k clusters. Only when r is chosen as a function of r_0  and then the pruning step in Algorithm 2 is performed do the properties seem to hold with k clusters. If this is the case, then I think the current wording in lines 66-70 is easily misinterpreted. Still, the stronger guarantee of uniform IP stability with r chosen as a function of r_0 is noteworthy.

- It seems that Algorithm 3 is simply the well-known algorithm of Single Link clustering. This should be labelled as such and cited.

**Questions:**

- A variant of Single-Link clustering is used in the work of Awasthi, Blum, and Sheffet (2012) to find optimal clusterings on perturbation stable instances. Given the algorithmic similarities, do you see any connections between these two notions of stability?

- Do you have any intuition as to why k-center approximation algorithms in particular are useful for IP-stability algorithms, given that IP-stability (in its original form) is based on average distances? While k-median and k-center are in general different notions, it is perhaps not clear a priori which objective could help inspire an algorithm for IP-stability. Have you considered the question of finding an algorithm that produces a clustering that is O(1)-IP stable and also constant approximate for other objectives?

**Limitations:**

Not applicable.

---

> ### Author Rebuttal · Authors · 2023-08-09
>
> We thank the reviewer for their comments. We address your concerns and questions below.
>
> > While Algorithms 1 and 2 are described very clearly, little motivation or intuition is given.
>
> Thanks for the suggestion. It did indeed require a good amount of work for us to figure out the details of the ball carving algorithm and we considered several simpler versions of the algorithm which didn’t work. As such, we realize that some motivation is in place. We will include this in the paper and provide some examples showing that various other natural approaches do not work.
>
> > A discussion of how Algorithms 1/2 differ from or build on the algorithms of Ahmadi et. al. would help highlight the novelty of the contributions
>
> Our algorithm is completely different from the algorithm of [Ahmadi et al] and is more sophisticated. Their O(n)-approximation relies on the standard metric embedding result to line metric, together with a simple greedy algorithm for IP-stability on line metric. The discussion of previous results is in line 51-55 of the paper.
>
> > The paragraph on lines 66-70 seems inconsistent with later claims
>
> You’re correct, Algorithm 1 works for any value of r but provides no guarantee on the number of clusters and we only get the property to hold with exactly k clusters for the right value of r which depends on r_0 (which in turn depends on P and k). We will make it clear in lines 66-70 that r depends on the set of points P and on k.
>
> > It seems that Algorithm 3 is simply the well-known algorithm of Single Link clustering.
>
> Yes, we will provide a citation appropriately.
>
> > A variant of Single-Link clustering is used in the work of Awasthi, Blum, and Sheffet (2012) to find optimal clusterings on perturbation stable instances. Given the algorithmic similarities, do you see any connections between these two notions of stability
>
> While the line of research on perturbation stability focuses on designing faster or more accurate algorithms utilizing the strong stability (and separability) conditions, here we aim to recover such stability conditions in general metrics approximability. We will expand on the discussion of stability in clustering in the related work section (line 158-165) and add comparison with the result of Awasthi, Blum, and Sheffet (2012).
>
> > Do you have any intuition as to why k-center approximation algorithms in particular are useful for IP-stability algorithms, given that IP-stability (in its original form) is based on average distances?
>
> k-center is particularly useful in the context of uniform approximate IP-stability. After getting said uniform property from the clustering output by Algorithm 1, it is natural to next consider the k-center algorithm. Indeed, for k-center, we have the relation that the minimum distance between clusters is also an upper bound on the distance from any point to its nearest center. This relation gives that when running Algorithm 2, we simultaneously get that each cluster is non-empty and the uniform upper bound on the diameter of each cluster in the final clustering. This property does not necessarily hold for k-median or k-means. We will add a comment on this in the paper.
>
> > Have you considered the question of finding an algorithm that produces a clustering that is O(1)-IP stable and also constant approximate for other objectives?
>
> Simultaneously achieving approximate IP stability as well as a global objective other than k-center is a very nice open question for future work (we also list it in Appendix D). We don’t see a way of directly extending our algorithm to k-median or k-means, for instance, because we use the diameter guarantee which is more specific to k-center.

---

> > ### Comment · Reviewer_kydQ · 2023-08-18
> >
> > I thank the authors for their detailed response. While the problem itself is fairly new as other reviewers point out, its formulation is natural and contributes to the growing literature on fairness and stability in clustering. The repurposing of existing algorithms is both subtle and clean, and the simultaneous guarantees for k-Center are appealing. Moreover, there are exciting follow-up questions for further research. I maintain my evaluation.

---

### Official Review · Reviewer_F6zy · 2023-07-07

**Soundness:** 4 excellent
**Presentation:** 3 good
**Contribution:** 3 good
**Rating:** 7
**Confidence:** 5

**Summary:**

This paper considers the problem of finding stable clusterings under a specific notion of stability - individual-preference stability which roughly requires the clustering produced to have the property where the average distance of any datapoint to points within its own cluster to be smaller than the average distance to points within any other cluster. The clustering is said to be $\alpha$-stable if the average in-cluster distance to be no more than a multiplicative $\alpha>1$ factor of the average distance to points assigned to any other cluster. This is a fairly natural problem, and was recently proposed by Ahmadi et. al. (ICML 22), who gave some preliminary results for it, including NP-Hardness of deciding whether the input dataset admits a $1$-stable clustering.

This paper makes three key contributions -

(a) It shows that given any dataset with $n$ points in a general metric space, and any desired number of clusters $k$, there always exists a  $k$-clustering that is $O(1)$-stable.

(b) This $O(1)$-stable $k$-clustering can be found by a computationally efficient algorithm. Moreover, the resulting clustering produced is a constant factor approximation to the optimal $k$-center objective.

(c) For min and max stability (i.e. where the average is replaced with min and max, respectively), they show that for any dataset in a general metric space, and for any choice of $k$, (i) a min-stable $k$-clustering always exists, and is achieved by the usual early-termination of Kruskals minimum spanning tree algorithm, and (ii) a 3 approximate max-stable $k$-clustering always exists and is achieved by the greedy $k$-center algorithm.

**Strengths:**

This paper considers a very natural question of obtaining a stable clustering, and substantially expands our understanding of this problem. Overall the paper is quite well-written and easy to read, and would be a well-suited for a venue such as NeurIPS.

(a) They show that given a set of any $n$ points in an arbitrary metric space, a $O(1)$-stable $k$-clustering always exists for any choice of $k$. Moreover, the clustering includes all $n$ points. The only known results prior to this work was the existence of a $1$-stable clustering for any set of $n$ points in a $1$-D Euclidean space or metrics induced by a weighted tree, or a bicriteria $O(\log^2 n/\epsilon)$ stable clustering achieved by discarding an $\epsilon$-fraction of the points from the input set. For the stricter requirement of clustering all $n$ points, the only known result was a trivial $O(n)$-stable clustering.

(b) The algorithm that finds this $O(1)$-stable $k$-clustering is simple and quite computationally efficient. The earlier bicriteria approximation result was achieved by applying the HST hammer naively. The new algorithm has the additional desirable property that the resulting clustering produced achieves a constant factor approximation for the $k$-center objective.


**Weaknesses:**

The paper has two main weaknesses, which in my opinion are not deal breakers given its other strengths.

(a) The extension to $f$-stable clusterings is obvious, and the results for $f=$ min and max stable clusterings are also very elementary. It would have been far more interesting had the authors characterized properties of a generic $f$ required such that any instance always admits an $O(1)$-$f$-stable $k$-clustering.

(b) The fact that the algorithm achieves a $O(1)$ approximation to the $k$-center objective seems like an afterthought. It doesnt seem like the algorithm was explicitly designed to additionally achieve good approximation guarantees for this other objective, and more like it happened to be that way.

**Questions:**

I understand that these questions are out of scope of this paper, but im curious about the following thing -

Given how you say that its only interesting to look at stability when the algorithm additionally gives good approximation guarantees for other natural clustering objective functions, I really want to know what happens when you simultaneously want to optimize for other objective functions such as say $k$-means or $k$-medians. Is it possible to get a clustering that achieves a good approximation for these standard clustering objectives, while simultaneously being approximately stable? Or are there cases where these two objectives of clustering "quality" and "stability" are at odds of each other - achieving one objective must necessarily incur a large penalty for the other objective?

My other question is related to your extension of the notion of stability, and one I have raised before in the weaknesses - can you characterize properties of a generic $f$ required such that any instance always admits an $O(1)$-$f$-stable $k$-clustering?

**Limitations:**

I dont see any obvious limitations of this work.

---

> ### Author Rebuttal · Authors · 2023-08-09
>
> We thank the reviewer for their comments. We address your concerns and questions below.
>
> > The extension to f-stable clusterings is obvious, and the results for min and max stable clusterings are also very elementary
>
> Indeed, the results for min and max stable clusterings come from quite straightforward observations, but we think it shows a nice connection between the individual preference objectives and classic algorithm problems (min spanning tree and k-center). The research question about generic f is a nice one for future work. It would be quite interesting to see if a general property holds that admits stable f-clusterings especially since the algorithms we have for min and max-stability use quite different techniques.
>
> > The fact that the algorithm achieves a O(1) approximation to the k-center objective seems like an afterthought
>
> From the point of view of our research process, this is correct: our main goal was to develop algorithms with small approximation factors for IP stability. The serendipity of the additional k-center guarantee doesn’t detract from the fact that simultaneously achieving individual and global clustering guarantees seems like it would be very valuable in practice (as we discussed in Section 1.2). A very nice open question (Appendix D and your question) is if good IP stability approximation can be combined with other global objectives.
>
> We believe our response above also answers your questions.

---

> > ### Comment · Reviewer_F6zy · 2023-08-21
> >
> > Thank you for your response. And as I said, I am already quite happy with this submission, and do think it should be accepted. My comments and questions were more a product of curiosity than anything. I do agree that trying to understand under what conditions can stability be combined with other objectives, and are there objectives that are inherently at odds with stability is an interesting question that one should pursue following this work. I am leaving my positive score unchanged.

---

### Official Review · Reviewer_zmPi · 2023-07-07

**Soundness:** 4 excellent
**Presentation:** 3 good
**Contribution:** 3 good
**Rating:** 7
**Confidence:** 3

**Summary:**

This paper concerns alpha-individual preference (IP) stable clusterings, meaning clusterings such that its average distance to points in its cluster is at most alpha times greater than to that in any other cluster. Previously, only O(n)-IP solutions were known, 1-IP solutions were known to not always exist, and we did not know if there existed O(1)-IP solutions. This paper answers affirmatively on metric spaces.

They provide an O(1)-IP algorithm on metric spaces which additionally is a constant-factor approximation to the k-center problem. The algorithm follows the popular greedy ball technique on metrics, which selects radius-bounded balls (bounded by O(1) times the optimal k-center solution) that cover many points and are distant from each other, and then fixes unassigned points (according to the greedy cluster order) and merges some clusters according to their closest k-center centers.

Additionally, they define variants Max-IP and Min-IP (where you instead look at the maximum/minimum distances, as opposed to the averages). They show Min-IP can be solved optimally by running Kruskal’s algorithm until there are k trees in the forest. Additionally, greedy k-center 3-approximates Max-IP.

Finally, the experimentally validate their IP algorithm on real and synthetic (adersarily-designed) datasets. On both, their algorithm outperforms k-means++, showing its robustness to hard instances.

**Strengths:**

This is a nice result that bridges a significant gap posed by previous works (i.e., the gap between O(n)-IP and O(1)-IP). In addition, it is complemented by the lower bound of 1-IP, and it implies a few nice future directions for this work, including finding the minimum alpha such that an alpha-IP solution exists and characterizing when 1-IP solutions exist. The extensions to Max-IP and Min-IP are also strong. For the most part, the writing was clear (though I did find some typos here and there). The methods seemed new in many ways, though they were heavily founded based off existing methods (i.e., clustering by balls on metric graphs).

**Weaknesses:**

The biggest weakness is that, while the improvement in approximation is significant, I think the problem is somewhat narrow in scope. I noticed the authors only cited one past work on this area (presumably, the one that proposed it), and the applications were not cited. While this does certainly seem like an interesting new problem, I am not sure of what its place is within ML research literature.

**Questions:**

1. Can you motivate this problem a bit more? Has anyone other than [1] tried attacking this problem?
2. Do you have any citations for uses?
3. I understand that IP was the main focus of the paper, but it would be nice to have an argument for the motivations of Min-IP and Max-IP. Do you know any applications they might work for?

**Limitations:**

I did not see a limitations section. It would be preferable but is not necessary for this work.

---

> ### Author Rebuttal · Authors · 2023-08-09
>
> We thank the reviewer for their comments. We address your concerns and questions below.
>
> > The biggest weakness is that, while the improvement in approximation is significant, I think the problem is somewhat narrow in scope. I noticed the authors only cited one past work on this area (presumably, the one that proposed it), and the applications were not cited. While this does certainly seem like an interesting new problem, I am not sure of what its place is within ML research literature.
>
> We thank the reviewer for their comments. It is true that individual preference stability is a new concept (though we would argue a rather natural criterion) that was only introduced in ICML’22. However, there are some connections to prior work that we will highlight.
>
> IP stable clustering is closely related to a burgeoning line of work on fairness in machine learning and, relatedly, on individual (rather than global) objectives. In a preliminary version of [1], they formulated the problem as a fair clustering problem (see [2]). Various notions of fairness in clustering have been examined [3, 4] (https://www.fairclustering.com, https://www.fairclustering.com/files/fair-clustering-taxonomy.pdf), and IP stability can be viewed as an alternate definition. We note that the notion of stability is more general than fairness (there may be other reasons to want stability, for example if the actors being clustered are strategic and may try to change clusters if unhappy).
>
> In terms of applications, to the best of our knowledge, we do not know any published work that uses this notion explicitly, but we do see potential applications. There are cases when we want to incorporate fairness in clustering, e.g., in bank loan, an applicant would be upset if their loan application gets rejected while another individual with similar features has an approved application. Similar applications can be argued for job listings, school applications, et cetera.
>
> Apart from [1], a somewhat similar notion was studied in [5]. In particular, in Sect. 1.1 of [5], it is mentioned that in general sense, for any good clustering notion, if $x \in C$ and $y \notin C$, then $x$ should be substantially closer to $C$ than $y$. [5] studied the case where the term ``substantially closer’’ is emphasized. Formally, a cluster $C$ is $(\alpha,\gamma)$-cluster if $P(x \in C)>\alpha$ and for any $y \notin C$, $d(y,C) \geq \gamma d(x,C)$. A $(\alpha, \gamma)$-clustering is a clustering where each cluster is an $(\alpha,\gamma)$-cluster. This problem becomes tractable when $\alpha > 0, \gamma >3$. Our case is corresponding to $(0,1)$-clustering. In fact, our algorithm finds a $(0, O(1))$-clustering.
>
> > I understand that IP was the main focus of the paper, but it would be nice to have an argument for the motivations of Min-IP and Max-IP. Do you know any applications they might work for?
>
> Similar to what the community does for fairness, we look into several notions of stability. While we do not know any applications yet, we believe that Min-IP, Max-IP, and more generally f-IP, are natural stability definitions.
>
>
> References:
>
> [1] Ahmadi, Saba, Pranjal Awasthi, Samir Khuller, Matthäus Kleindessner, Jamie Morgenstern, Pattara Sukprasert, and Ali Vakilian. "Individual Preference Stability for Clustering." In International Conference on Machine Learning. 2022.
>
> [2] Kleindessner, Matthäus, Pranjal Awasthi, and Jamie Morgenstern. "A notion of individual fairness for clustering." arXiv preprint arXiv:2006.04960 (2020).
>
> [3] Brubach, Brian, Deeparnab Chakrabarty, John P. Dickerson, Seyed Esmaeili, Matthäus Kleindessner, Marina Knittel, Jamie Morgenstern, Samira Samadi, Aravind Srinivasan, and Leonidas Tsepenekas. "Fairness in Clustering."
>
> [4] Chhabra, Anshuman, Karina Masalkovaitė, and Prasant Mohapatra. "An overview of fairness in clustering." IEEE Access 9 (2021): 130698-130720.
>
> [5] Daniely, Amit, Nati Linial, and Michael Saks. "Clustering is difficult only when it does not matter." arXiv preprint arXiv:1205.4891 (2012).

---

### Official Review · Reviewer_d1bb · 2023-07-11

**Soundness:** 4 excellent
**Presentation:** 4 excellent
**Contribution:** 4 excellent
**Rating:** 7
**Confidence:** 5

**Summary:**

In $\alpha$-individual preference (IP) stable clustering, the average distance between every data point to other points in its cluster must be at most $\alpha$ times the average distance between it to points in any other cluster.

This paper gives the first polynomial time $O(1)$-IP stable clustering algorithm in general metrics, which improves [Ahmadi et.al, ICML 2022] 's $O(n)$-IP stable result. The algorithm in this paper also has more interesting features that I appreciate. Firstly, it satisfies an even stronger uniform IP stability guarantee. Precisely, there exists a global parameter $r$, such that for every data point, its average distance to points in its cluster is bounded by $O(r)$ while its average distance to point in any other cluster is at least $O(r)$. Secondly, the k-clustering it finds is also a constant factor approximation to the classical k-center clustering problem, which makes the solution much more meaningful in applications. I think this result shows that k-center clustering can admit IP stability without paying too much. Exciting news!

Moreover, the algorithm presented is clean and easy to follow with its analysis building on certain clever observations. Authors also do experiments against [Ahmadi et.al, ICML 2022]  to show the practical value of their algorithm. At last, the authors also study other definitions of IP-stable and obtain similar results.

Overall, I recommend "accept".

**Strengths:**

1. The first efficient algorithm to compute O(1)-IP stable approximate clustering in general metrics. Additionally, the algorithm has more important features beyond IP stability. The result significantly improves the previous works.

2. The algorithm and analysis are interesting and have some non-trivial insights.

3. The experiments also outperform the baseline.

**Weaknesses:**

Both the constants for IP approximation and k-center approximation may be too large. While IP stability is motivated by practical applications, I would expect that the guarantee is not a quite large number such as 2 or 3. The paper's experiment already shows that the worst case may be avoided in practice and I would like to see more discussion in this direction. Anyway, as a theoretical submission, I think this is totally fine and does not lower my rating.

**Questions:**

1. It is argued that the running time can be improved in Euclidean space. What is the exact running time that you can achieve? What about $\tilde{O}(nkd)$? (a typical running time for Euclidean $k$-clustering.)

2. Have you formally proved that your algorithm returns an $O(1)$-approximation for $k$-center in the main text? I know it can be seen from Algorithm 2 but it is better to provide formal proof and write the exact approximation ratio.

**Limitations:**

I do not see the potential negative societal impact.

---

> ### Author Rebuttal · Authors · 2023-08-09
>
> We thank the reviewer for their comments. Below is our response.
>
> > "[...] the constants for IP approximation and k-center approximation may be too large."
>
> The focus of this paper is to give a constant factor approximation, improving upon the only O(n)-approximation of the IP-stability clustering problem. For the sake of clarity of the presentation, we did not try to optimize the constants here.
>
> > "The paper's experiment already shows that the worst case may be avoided in practice and I would like to see more discussion in this direction."
>
> This is indeed an interesting future direction to explore the approximability of IP-stability for instances in practice. A challenge in this direction is to identify the properties that result in better IP-stable clustering guarantees.
>
> > It is argued that the running time can be improved in Euclidean space. What is the exact running time that you can achieve? What about O~(ndk) (a typical running time for Euclidean k-clustering.)?
>
> We have opted for simplicity and generality to ensure our algorithm applies to any metric space. As stated in Section 3, our algorithm can be naively implemented in $\tilde{O}(n^2T)$ time for any metric space where $T$ is the time to compute a distance between two points. However, we believe that for specialized metrics, such as Euclidean metrics, it is an interesting future question to obtain faster algorithms. For example, one can show that for *constant* dimensional Euclidean space, our algorithm can be made to run in O(nk) time.
>
> > Have you formally proved that your algorithm returns an O(1) approximation for k-center in the main text? I know it can be seen from Algorithm 2 but it is better to provide formal proof and write the exact approximation ratio
>
> The k-center guarantee is shown in Theorem 3.1. Per the reviewer's suggestion, we will include the exact k-center approximation factor in the final version.

---

### Decision · Program_Chairs · 2023-09-21

**Decision:**

Accept (spotlight)

**Comment:**

All of the reviewers were positive about this paper. The paper considers an interesting problem recently introduced, individually stable clustering. The paper offers interesting new algorithmic insights into the problem.